# Factors associated with the very high caesarean section rate in urban areas of Vietnam

**Hoang Thi Nam Giang**[1]*, **Do Thi Thuy Duy**[1], **Le Tho Minh Hieu**[1], **Nguyen Lam Vuong**[2], **Nguyen Thi Tu Ngoc**[3], **Mai Thi Phuong**[4], **Nguyen Tien Huy**[5]

1 School of Medicine and Pharmacy, The University of Da Nang, Da Nang, Vietnam, 2 University of Medicine and Pharmacy at Ho Chi Minh City, Ho Chi Minh City, Vietnam, 3 Thai Nguyen University of Medicine and Pharmacy, Thai Nguyen, Vietnam, 4 Hung Vuong Hospital, Ho Chi Minh City, Vietnam, 5 School of Tropical Medicine and Global Health, Nagasaki University, Nagasaki, Japan

* htngiang@ud.edu.vn

## Abstract

### Background

Caesarean section and associated factors require detailed investigation globally. This study aims to determine the rate and associated factors of caesarean deliveries in urban areas of Vietnam.

### Methods

A cross-sectional study using questionnaire answered by women who had infants aged under 30 months was conducted from March to May 2021. Data were collected in 18 commune health centres in two cities during the day of routine immunization. Multivariable logistic regression was performed to assess factors associated with caesarean section.

### Results

The overall caesarean section rate was 49.6%. The caesarean section rate in private hospitals (57.8%) were significantly higher than in public hospitals (49.1%). Caesarean section rate in first-time mothers (47.1%) were as high as this rate among mothers who had given birth before (50.6%). Factors associated with higher rate of caesarean section include increasing in women's age, pre-pregnancy body mass index, gestational weight gain, and infant's birth weight; first-time mothers; mothers living in urban areas; and mothers giving birth in private hospitals.

### Conclusions

This study revealed a high rate of caesarean deliveries in urban areas of Vietnam. Comprehensive investigations of both medical and non-medical reasons for caesarean deliveries in Vietnam are urgent needs to shape the prioritized interventions.

**Data Availability Statement:** Data are available on Figshare (DOI: https://doi.org/10.6084/m9.figshare.20416581.v1).

**Funding:** This research is funded by Funds for Science and Technology Development of the

University of Danang under project number B2020-DN01-28. The funders had no role in study design, data collection and analysis, decision to publish, or preparation of the manuscript.

**Competing interests:** The authors have declared that no competing interests exist.

## Introduction

Vaginal birth is a normal physiological process. Caesarean section is a medical technology that could save mother's and newborn's lives when there are medical reasons. Based on the latest available data, the Statement on Caesarean Section Rates by the World Health Organization (WHO) emphasized that a caesarean section rate of higher than 10% at population level was not associated with lower maternal and neonatal mortality rate [1]. However, the caesarean section rate is continuing to increase globally irrespective of income level [1]. A systematic review and ecological analysis from 154 countries shows that more than 1 in 5 women gave birth by caesarean globally and the greatest increase was observed in countries of Asia Region [2].

Vietnam is a lower middle-income country in South-eastern Asia and the world's fifteenth-most populous country. Over the last 20 years, the caesarean section rate in Vietnam has increased dramatically, from 3.4% in 1994 to 27.5% in 2014 [2]. The caesarean section rate in rural areas of Vietnam was 21% in 2014, whereas this rate in urban areas was 43% [3]. Current studies even reported caesarean section rates of higher than 50% in an urban area [4]. Although the caesarean section rate has increased dramatically, there is a paucity of study investigating factors associated with caesarean section in Vietnam [4–6]. Even the available data on caesarean section rate in Vietnam is mainly from the report of The United Nations Children's Fund (UNICEF).

Multiple factors and interactions contribute to the caesarean deliveries. These reasons include but not limited to women and families' preferences, convenience, financial incentives, lack of regulation, lack of supervision and training on vaginal delivery, the views and beliefs of health care professionals, maternal educational level, previous caesarean section, type of hospital as private or public, socioeconomic status [7, 8]. Understanding current caesarean section rate in Vietnam and associated factors is important to reduce unnecessary use of the caesarean section. Therefore, the current study aims to identify the caesarean section rate in urban areas of Vietnam and associated factors. Understanding these factors may provide S1 Appendix for future research and shaping policy on maternal healthcare services.

## Methods

### Study settings

In recent years, Vietnam has made impressive progress in economic and social development and was classified as a lower middle-income country in 2010. The country is divided administratively into three regions including Northern, Central, and Southern Vietnam. In 2021, the average population of Vietnam reached more than 98.5 million people. The total fertility rate is 2 births per woman in 2019. The healthcare system of the country consists of both public and private sectors. The public sector comprise four administrative levels: national/central hospital, provincial hospital, district hospital and commune health centre. In Vietnam, about 92% of deliveries take place in health facilities [9].

In this study, data were collected from two cities of Vietnam: Thai Nguyen and Da Nang. Thai Nguyen is the ninth largest city of Vietnam, which is located in the North. The city has a population of 420,000 people in 2018 and includes 21 urban communes and 11 rural communes. Da Nang is the fifth largest city in Vietnam, which is located in the Central Region. The city includes 45 urban communes and 11 rural communes.

### Study design and inclusion criteria

From March to May 2021, a cross-sectional survey using a printed questionnaire answered by women who had children aged under 30 months was conducted in two cities in Vietnam to

investigate breastfeeding patterns. Using data from this study, we aimed to identify the caesarean section rate and factors associated with increased caesarean deliveries.

Mothers having children aged under 30 months and singleton mothers were selected to the study.

## Selecting study participants

The Expanded Programme on Immunization in Vietnam provides immunization to 10 vaccine preventable diseases free of charge. Infants receive vaccines in community health centres in fixed facility sessions from 2 to 3 days per month. All mother, whose children were vaccinated in 18 commune health centres on the fixed date of each commune health centre, were invited to participate in the study. All eligible mothers were invited until the desired number of participants had been recruited. One research team member was responsible for managing the data collection in each city.

## Sample size calculation

Regarding sample size calculation, we assumed the true prevalence of exclusive breastfeeding during hospital stay is 33.0% and required the estimate to be within 0.022 of the true value within the 95% confidence interval, with a 5% non-response rate. Based on these assumptions, the minimum sample size was calculated to be 1843 mothers.

## Questionnaire

The questionnaire consists of socio-demographic characteristics including maternal age, educational level (primary school, secondary school, high school, intermediate degree, university/postgraduate), place of residence (urban, rural), average family income (million VND), parity, complication during pregnancy and at birth, mode of delivery (caesarean section or vaginal birth), level of hospital of birth (central/provincial, district, private, commune), pre-pregnancy body mass index (BMI), gestational weight gain (kg), infant's sex and birth weight. Living in urban areas or rural areas was based on the administrative definition of the communes.

## Data quality assurance

All data collectors were trained by member of research team before collecting data. A pre-test of the questionnaire was done on 10 mothers. After collecting comments from those mothers, the questionnaire was adjusted to improve the clarity and understanding.

## Data analysis

Each filled questionnaire was manually checked for errors and entered into EpiData version 3.1 for analysis. Summary statistics were done for all variables using frequency and percentage for categorical variables, and mean and standard deviation or median and interquartile range for continuous variables. Binary logistic regression was used to examine the association between maternal and infant characteristics with caesarean section. The crude odds ratio (OR) and 95% confidence interval (95% CI) were reported. Thereafter, a multivariable logistic regression with complete-case analysis included all variables in binary logistic regression was performed. Maternal and infant characteristics were included in multivariable logistic regression are maternal age, educational level, place of residence, average family income, parity, complication during pregnancy and at birth, mode of delivery, pre-pregnancy BMI, gestational weight gain (kg), infant's sex, and infant's birth weight. The adjusted OR (aOR) and 95% CI were reported. Chi-squared test was used for comparison of two proportions. A p-value of less

than 0.05 was considered as statistically significant. Analyses were conducted in the statistical software R version 3.6.1. Our study followed A Consensus-Based Checklist for Reporting of Survey Studies (CROSS) [10].

### Ethical approval and informed consent

The study was conducted according to the guidelines of the Declaration of Helsinki, and approved by the Institutional Review Board of University of Medicine and Pharmacy at Ho Chi Minh City according to Decision No. 990/HĐĐĐ-ĐHYD date 11/01/2020. The written informed consent was obtained from all the participants.

## Results

### Description of study participants

Among 1812 mothers included in the present analysis, 29.5% of mothers having birth for the first time, mean pre-pregnancy BMI of mothers was $20.4 \pm 2.4$ kg/m$^2$ and mean maternal weight gain during pregnancy was $12.4 \pm 4.7$ kg. There were 64.7% of the mothers giving birth in central/provincial hospital, 26.7% in district hospital and 8.2% in private hospital. Description of maternal and infant characteristics are presented in Table 1.

**Table 1. Maternal and infant characteristics.**

| Characteristics | n | Summary statistics (N = 1812) |
|---|---|---|
| **Maternal characteristics** | | |
| Age (years) | 1801 | 29.7±5.1 |
| Education qualification | 1805 | |
| • Primary/Secondary | | 277 (15.4) |
| • High school/intermediate degree | | 982 (54.4) |
| • University/postgraduate | | 546 (30.2) |
| Complication during pregnancy and at birth | 1810 | 170 (9.4) |
| Place of residence classified as rural | 1749 | 801 (45.8) |
| Primiparous | 1808 | 534 (29.5) |
| Caesarean section | 1808 | 896 (49.6) |
| Hospital of birth | 1799 | |
| • Central/Provincial | | 1164 (64.7) |
| • District | | 480 (26.7) |
| • Private | | 147 (8.2) |
| • Commune | | 8 (0.4) |
| Family monthly income (million VND) | 1720 | 12.2±6.4 |
| Pre-pregnancy BMI (kg/m$^2$) | 1776 | 20.4±2.4 |
| Gestational weight gain (kg) | 1800 | 12.4±4.7 |
| **Infant characteristics** | | |
| Sex, male | 1806 | 929 (51.4) |
| Birth weight (kg) | 1810 | 3.2±0.4 |

Summary statistics are mean ± SD for continuous variables, and frequency (%) for categorical variables. BMI: body mass index; SD: standard deviation; VND: Vietnam dong.

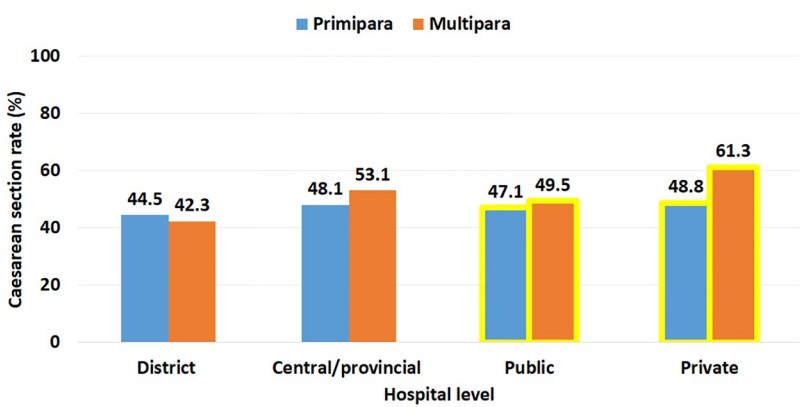

**Fig 1. Caesarean section rate by level of hospitals and maternal parity, Vietnam, 2022.**

### Caesarean section rate

The current study shows that 49.6% of women gave birth by caesarean section. The caesarean section rate of women who gave birth in private hospitals (n = 85, 57.8%) was significantly higher than in public hospital (n = 806, 49.1%) (p = 0.04). There were 251 (47.1%) of primipara women gave birth by caesarean section and 644 (50.6%) of multipara women gave birth by caesarean section. The difference was not significant, p = 0.175. Among 206 women (42.9%) gave birth by caesarean section in district hospital, the caesarean section rate was higher in primipara women than in multipara women (p = 0.660). Detail on caesarean section rate by level of hospitals and maternal parity are shown in Fig 1.

### Factors associated with higher rate of caesarean section

In the multivariable logistic regression analysis, after adjusting for maternal and infant characteristics in Table 2, age of mothers, place of resident, parity, pre-pregnancy BMI, gestational weight gain, hospital of birth and infant's birth weight were significantly associated with an increased risk of caesarean section. For one-year increase in age of mothers, the odds of caesarean section increases by 1.07 times (95% CI: 1.04 to 1.09). The odds of caesarean section in mothers who were in residence at urban areas was 1.25 times higher than in mothers who were in residence at rural areas (95% CI: 1.01 to 1.55). First-time mothers were more likely to have a caesarean section than mothers who had given birth before (aOR: 1.38, 95% CI: 1.06 to 1.79). Women with higher pre-pregnancy BMI was also associated with an increased probability of caesarean section (aOR: 1.08, 95% CI: 1.04 to 1.13). For one-kg increase in weight gain, the odds of caesarean section increased by 1.04 times (95% CI: 1.02 to 1.07). In private hospitals, women were 1.6 times more likely to undergo a caesarean delivery than who delivered in public systems (95% CI: 1.09 to 2.37).

### Discussion

This cross-sectional study found that the caesarean section rate was 49.6%. Caesarean section rate among women giving birth in private hospitals (57.8%) was significantly higher than in public hospitals (49.1%). The rate of caesarean section among first-time mothers was 47.1% and among mothers who had given birth before was 50.6%. Factors associated with higher rate of caesarean section includes increasing in women' age, pre-pregnancy BMI, gestational

**Table 2. Factors associated with caesarean section in Vietnam, 2022.**

| Characteristics | Vaginal birth | Caesarean section | Unadjusted OR (95% CI) | P value | Adjusted OR (95% CI) | P value |
|---|---|---|---|---|---|---|
| **Age of mother** | 28.9 ± 4.9 | 30.5 ± 5.2 | 1.06 (1.04 to 1.08) | < 0.001 | 1.07 (1.04 to 1.09) | < 0.001 |
| **Mother education level** | | | | | | |
| • Primary & Secondary | 140 (50.7) | 136 (49.3) | Reference | | Reference | |
| • High school/ Intermediate degree | 514 (52.4) | 466 (47.6) | 0.93 (0.71 to 1.22) | 0.613 | 0.93 (0.69 to 1.27) | 0.652 |
| • University/ Postgraduate | 254 (46.5) | 292 (53.5) | 1.18 (0.89 to 1.58) | 0.255 | 1.03 (0.73 to 1.46) | 0.862 |
| **Family monthly income (million VND)** | 12.2 ± 6.6 | 12.2 ± 6.2 | 1.00 (0.98 to 1.01) | 0.973 | 0.99 (0.97 to 1.01) | 0.598 |
| **Place of residence** | | | | | | |
| • Rural | 444 (55.5) | 356 (44.5) | Reference | | Reference | |
| • Urban | 444 (46.9) | 502 (53.1) | 1.41 (1.17 to 1.70) | < 0.001 | 1.25 (1.01 to 1.55) | 0.042 |
| **Parity** | | | | | | |
| • Multipara | 628 (49.4) | 644 (50.6) | Reference | | Reference | |
| • Primipara | 282 (52.9) | 251 (47.1) | 0.87 (0.71 to 1.06) | 0.170 | 1.38 (1.06 to 1.79) | 0.016 |
| **Complication during pregnancy and at birth** | | | | | | |
| • No | 841 (51.3) | 797 (48.7) | Reference | | Reference | |
| • Yes | 71 (42.0) | 98 (58.0) | 1.46 (1.06 to 2.01) | 0.021 | 1.19 (0.84 to 1.7) | 0.325 |
| **Pre-pregnancy BMI** | 20.2 ± 2.2 | 20.7 ± 2.5 | 1.1 (1.06 to 1.14) | < 0.001 | 1.08 (1.04 to 1.13) | < 0.001 |
| **Gestational weight gain (kg)** | 11.9 ± 4.6 | 12.8 ± 4.8 | 1.04 (1.02 to 1.06) | < 0.001 | 1.04 (1.02 to 1.07) | < 0.001 |
| **Hospital of birth** | | | | | | |
| • Public | 835 (50.9) | 806 (59.1) | Reference | | Reference | |
| • Private | 62 (42.2) | 85 (57.8) | 1.42 (1.01 to 2.00) | 0.043 | 1.60 (1.09 to 2.37) | 0.017 |
| **Infant's sex** | | | | | | |
| • Female | 460 (52.5) | 416 (47.5) | Reference | | Reference | |
| • Male | 448 (48.3) | 479 (51.7) | 1.18 (0.98 to 1.42) | 0.076 | 1.20 (0.98 to 1.47) | 0.084 |
| **Infant's birth weight (kg)** | 3.2 ± 0.4 | 3.3 ± 0.4 | 1.79 (1.42 to 2.26) | < 0.001 | 1.31 (1.01 to 1.69) | 0.039 |

Summary statistics are mean ± SD for continuous variables, and frequency (%) for categorical variables. BMI, body mass index; SD, standard deviation; VND, Vietnam dong.

weight gain, and infant's birth weight; first-time mothers; mothers who were resident in urban areas; mothers who gave birth in private hospitals.

The overall caesarean section rate of 49.6% was considerably higher than the caesarean section rate at population level of 10% recommended by WHO [1]. The caesarean section rate in current study is also higher than the rate of caesarean section rate of 27.5% in Vietnam from 2011 to 2014 reported by UNICEF [2] and average caesarean section rate of 15.9% in South-eastern Asia [2]. However, this rate is consistent with the rate reported in urban areas in Vietnam of study 2018 (58.6%) [4] and 2020 (44.3%) [11]. Our study was conducted in two first class cities in Vietnam and both cities are among the ten largest cities in Vietnam. Even in the city, mothers who were in residence at urban communes were 1.25 times more likely to undergo caesarean section than mothers who were in residence at rural communes. This trend is also observed in previous study [12–14]. According to UNICEF, the caesarean section rate in urban areas of Vietnam has steadily increased from 12% in 1997 to 23% in 2002, 31% in 2010, and 43% in 2014 [3, 11]. On the other hand, the likelihood of caesarean section increases with the increase of wealth [15]. Therefore, the high rate of caesarean section in our current study seems to be consistent with trends and projections of caesarean section rate in urban areas.

Caesarean section rate increased with advancing maternal age. Our finding is consistent with studies that have been carried out in Vietnam and other countries [11, 16, 17]. The risk of pregnancy complications and adverse outcomes including chronic hypertension, diabetes, primary caesarean, excessive labour bleeding, and pregnancy hypertension is associated with increasing maternal age (women aged ≥35) [17]. There was a very high rate of 47.1% primipara women in our study giving birth by caesarean section compared with finding from other countries [18, 19]. In comparison with multipara women, primipara women had 1.38 the odds for caesarean section. The reasons for increasing caesarean rate are multifactorial including both clinical indications and non-clinical factors. Health care provider factors consist of obstetrician's ability to schedule the caesarean delivery at their convenience, financial incentive, shorter duration of a caesarean delivery compared to a vaginal birth, lack of training of obstetricians in vaginal delivery seem to be important factors influencing caesarean section rate in hospital [20]. Another non-medical factors contributing to the rise of caesarean section rate could be elective caesarean sections on maternal request and maternal fear of delivery [21]. These non-medical reasons could play important roles in increasing caesarean section rate in primipara women. Thereafter, previous caesarean section becomes the main medical indication in subsequent pregnancies and contributes significantly to the rise of caesarean section rate [4, 22, 23]. Therefore, reducing caesarean delivery among first-time mothers will reduce caesarean section among multipara women in subsequent pregnancies.

In this study conducted among Vietnamese women, we also found that increasing pre-pregnancy BMI was independently associated with increased odds of caesarean section. There have been numerous studies that have concluded the link between obesity and caesarean section [24–26]. In obese women, the pregnancy complications and adverse outcomes such as hypertensive disorders of pregnancy and macrosomia may result in a higher likelihood of caesarean section [27–29]. In addition, increased gestational weight gain was linked to elevated risk of caesarean section. Our findings in accordance with the results of previous studies [24, 30]. The risk of fetal distress, low Apgar score, and large for gestational age are higher in women gained excessive weight during pregnancy compared with women in the normal weight gain [27, 31, 32]. These may result in a higher need for a caesarean section. Prevention of overweight and obesity prior pregnancy and excess weight gain during pregnancy should be included in strategies to reduce caesarean section.

## Limitations

This study has several limitations. First, as this study was conducted in two cities with very high urbanization ratio, the caesarean section rate may not be a population-representative level in Vietnam. Second, the estimated rate of caesarean section in private hospitals may be less precise than in the public hospitals due to difference in the number of participants included. Third, investigating caesarean section rate and associated factors are not the primary objectives of this study; therefore, we do not comprehensively investigate medical and non-medical indicators of caesarean section. Fourth, pre-pregnancy BMI calculation in our study was based on the self-reporting of maternal height and weight. This may lead to the misclassification of BMI as reported in numerous epidemiological studies [25]. Nevertheless, our findings highlight a very high caesarean section rate of about 50% in both primiparas and multiparas and provide useful insights into the influential factors on caesarean section.

## Conclusion

The overall caesarean section rate among the women who brought their children to the commune health centres for routine vaccination is particularly high at 49.6%. The rate of caesarean

section in primiparas was 47.1% and in multiparas was 50.6%. Current findings show that after controlling for maternal and infant's characteristics, living in urban areas, giving birth in private hospitals, increasing in maternal age, pre-pregnancy BMI, gestational weight gain, and infant's birth weight associated with increased the likelihood of undergoing a caesarean section. Comprehensive investigations of both medical and non-medical reasons for caesarean deliveries in Vietnam are urgent needs to shape the prioritized interventions.

## Supporting information

**S1 Appendix. Checklist for Reporting of Survey Studies (CROSS).**
(DOCX)

## Author Contributions

**Conceptualization:** Hoang Thi Nam Giang.

**Data curation:** Hoang Thi Nam Giang, Do Thi Thuy Duy, Le Tho Minh Hieu.

**Formal analysis:** Hoang Thi Nam Giang.

**Funding acquisition:** Hoang Thi Nam Giang.

**Investigation:** Hoang Thi Nam Giang, Do Thi Thuy Duy, Le Tho Minh Hieu, Nguyen Lam Vuong, Nguyen Thi Tu Ngoc, Mai Thi Phuong.

**Methodology:** Hoang Thi Nam Giang.

**Project administration:** Hoang Thi Nam Giang.

**Resources:** Hoang Thi Nam Giang.

**Software:** Hoang Thi Nam Giang.

**Supervision:** Hoang Thi Nam Giang.

**Validation:** Hoang Thi Nam Giang, Nguyen Tien Huy.

**Visualization:** Hoang Thi Nam Giang.

**Writing – original draft:** Hoang Thi Nam Giang.

**Writing – review & editing:** Hoang Thi Nam Giang, Do Thi Thuy Duy, Le Tho Minh Hieu, Nguyen Lam Vuong, Nguyen Thi Tu Ngoc, Mai Thi Phuong, Nguyen Tien Huy.

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
