## [Decision Letter · Decision Letter 0]

18 Jul 2022

PONE-D-22-16814Factors associated with the very high caesarean section rate in urban areas of VietnamPLOS ONE

Dear Dr. Hoang Thi Nam Giang,

Thank you for submitting your manuscript to PLOS ONE. After careful consideration, we feel that it has great contribution for the quality improvement in the area, especially in Vietnam, however, does not fully meet PLOS ONE’s publication criteria as it currently stands. Therefore, we invite you to submit a revised version of the manuscript that addresses the points raised during the review process.

We look forward to receiving your revised manuscript.

Kind regards,

Takashi Watari, MD, MHQS, Ph.D

Academic Editor

PLOS ONE

Journal Requirements:

Additional Editor Comments:

Thanks for the opportunity to review this manuscript. It was very well-written and the limitations are clearly described. It drives interesting evidences on the uprising of cesarean section in Vietnan, and I think the results presented would be similar in other countries with similar development. I only suggest that maternal and infant characteristics that were considered for adjustment are not described, neither in methods nor in table 2. Consider including it.

Reviewers' comments:

Reviewer's Responses to Questions

**Comments to the Author**

1. Is the manuscript technically sound, and do the data support the conclusions?

Reviewer #1: Yes

2. Has the statistical analysis been performed appropriately and rigorously? 

Reviewer #1: Yes

3. Have the authors made all data underlying the findings in their manuscript fully available?

Reviewer #1: Yes

4. Is the manuscript presented in an intelligible fashion and written in standard English?

Reviewer #1: Yes

5. Review Comments to the Author

Reviewer #1: Dear editor,

Thanks for the opportunity to review this manuscript. It was very well-written and the limitations are clearly described. It drives interesting evidences on the uprising of cesarean section in Vietnan, and I think the results presented would be similar in other countries with similar development. I only suggest that maternal and infant characteristics that were considered for adjustment are not described, neither in methods nor in table 2. Consider including it.

6. PLOS authors have the option to publish the peer review history of their article (what does this mean?). If published, this will include your full peer review and any attached files.

Reviewer #1: **Yes: **Jose Paulo Siqueira Guida

---

## [Author Response · Author response to Decision Letter 0]

2 Aug 2022

Dear Editorial Board Member

PLOS ONE

Re: Factors associated with the very high caesarean section rate in urban areas of Vietnam

Thank you very much for the opportunity to revise this manuscript. We have addressed the comments in the point-by-point response. 

We hope that our responses are satisfactory to you.

Yours sincerely,

Hoang Thi Nam Giang

Reviewer’s comments

Comment: Thanks for the opportunity to review this manuscript. It was very well-written and the limitations are clearly described. It drives interesting evidences on the uprising of cesarean section in Vietnan, and I think the results presented would be similar in other countries with similar development. I only suggest that maternal and infant characteristics that were considered for adjustment are not described, neither in methods nor in table 2. Consider including it.

Response: Thanks for your comments. We described maternal and infant characteristics that were considered for adjustment in methods and in Table 2 as follow: 

Page 6, line 128 and 132: “Maternal and infant characteristics were included in multivariable logistic regression are maternal age, educational level, place of residence, average family income, parity, complication during pregnancy and at birth, mode of delivery, pre-pregnancy body mass index (BMI), gestational weight gain (kg), infant’s sex, and infant’s birth weight”

Thank you for your comments.

---

## [Editor Report · Decision Letter 1]

17 Aug 2022

Factors associated with the very high caesarean section rate in urban areas of Vietnam

PONE-D-22-16814R1

Dear Dr. Hoang Giang

We’re pleased to inform you that your manuscript has been judged scientifically suitable for publication and will be formally accepted for publication once it meets all outstanding technical requirements.

Kind regards,

Takashi Watari, MD, MQHS, Ph.D

Academic Editor

PLOS ONE

Additional Editor Comments (optional):　

I believe that this research is extremely effective in improving the quality of healthcare in Vietnam. I wish you all the best in your research field.

---

## [Editor Report · Acceptance letter]

19 Aug 2022

PONE-D-22-16814R1 

Factors associated with the very high caesarean section rate in urban areas of Vietnam 

Dear Dr. Giang:

I'm pleased to inform you that your manuscript has been deemed suitable for publication in PLOS ONE. Congratulations! Your manuscript is now with our production department. 

Kind regards, 

on behalf of

Dr. Takashi Watari 

Academic Editor

PLOS ONE